# All-optical structuring of laser-driven proton beam profiles

Lieselotte Obst-Huebl [1,2], Tim Ziegler [1,2], Florian-Emanuel Brack [1,2], João Branco[1,2], Michael Bussmann[1], Thomas E. Cowan [1,2], Chandra B. Curry [3,4], Frederico Fiuza [3], Marco Garten [1,2], Maxence Gauthier[3], Sebastian Göde[5], Siegfried H. Glenzer[3], Axel Huebl [1,2], Arie Irman[1], Jongjin B. Kim [3], Thomas Kluge [1], Stephan D. Kraft [1], Florian Kroll[1], Josefine Metzkes-Ng[1], Richard Pausch [1,2], Irene Prencipe[1], Martin Rehwald[1,2], Christian Roedel [6], Hans-Peter Schlenvoigt [1], Ulrich Schramm [1,2] & Karl Zeil[1]

Extreme field gradients intrinsic to relativistic laser-interactions with thin solid targets enable compact MeV proton accelerators with unique bunch characteristics. Yet, direct control of the proton beam profile is usually not possible. Here we present a readily applicable all-optical approach to imprint detailed spatial information from the driving laser pulse onto the proton bunch. In a series of experiments, counter-intuitively, the spatial profile of the energetic proton bunch was found to exhibit identical structures as the fraction of the laser pulse passing around a target of limited size. Such information transfer between the laser pulse and the naturally delayed proton bunch is attributed to the formation of quasi-static electric fields in the beam path by ionization of residual gas. Essentially acting as a programmable memory, these fields provide access to a higher level of proton beam manipulation.

[1] Helmholtz-Zentrum Dresden - Rossendorf, Institute of Radiation Physics, Bautzner Landstr. 400, 01328 Dresden, Germany. [2] Technische Universität Dresden, 01062 Dresden, Germany. [3] High Energy Density Science Division, SLAC National Accelerator Laboratory, Menlo Park, CA, 94025 USA. [4] University of Alberta, Edmonton, Alberta, T6G 1H9 Canada. [5] European XFEL GmbH, Holzkoppel 4, 22869 Schenefeld, Germany. [6] Helmholtz Institute Jena, Fröbelstieg 3, 07743 Jena, Germany. Correspondence and requests for materials should be addressed to L.O-H. (email: l.obst-huebl@hzdr.de)

Laser-plasma proton accelerators[1] have attracted attention in a wide range of scientific and applied areas, as they represent compact sources of intense and energetic[2,3] proton bunches with unique spatial and spatio-temporal qualities, namely, point source characteristic and sub-ps bunch duration[4]. Time-resolved radiography[5] of transient plasma phenomena with internal[6] and external[7–9] proton probes, materials and warm dense matter research[10,11], archeological surveys[12], high dose-rate radiobiology[13] and translational research in radiation oncology[14] only highlight a few. Many of these applications rely on tailored transport and beam shaping techniques for the laser-accelerated proton beam[15–17] that are derived from conventional approaches and suffer from low efficiency. More effective control of the beam however is complicated as, close to the source, it requires electromagnetic fields exceeding the extreme field strengths in the order of MV $\mu m^{-1}$ inherent to the acceleration process.

Laser-proton acceleration[1,18,19] relies on the transfer of energy from a high power laser pulse tightly focused at the surface of a typically opaque target to the target electrons, which immediately reach relativistic energies. Part of this energy is transferred to protons originating from the target surfaces in a plasma expansion dominantly perpendicular to the target surfaces. This mechanism, known as target normal sheath acceleration (TNSA)[20,21], being the most robust scheme for laser-proton acceleration and studied most widely for a variety of laser and target parameters[1,21–23], allows for direct manipulation of collective beam parameters like pointing and divergence. Influencing the symmetry of the accelerating electron sheath has been demonstrated by shaping of the focal spot[24,25], by introducing a laser pulse front tilt[26], or by micro-engineering of the target surface[4,27–29]. The limitation of the lateral target size (so-called mass-limited targets with few 10 μm in diameter) has been pursued as a complementary approach confining the electron sheath. Generally motivated by the pursuit of increasing proton energies[30–33], additional improvements of the beam divergence were reported[32] as well as an influence on proton beam confinement by laser light leaking around the target[33]. These approaches, while key to control proton beam propagation, are often difficult to realize in application-oriented experiments and, more importantly, do not allow tailoring of structures on the generated beam profile according to a specific design. Particularly, blocking dose in certain areas across the proton beam could be of significant interest for those applications where inserting a metal mask into the particle beam as an alternative beam structuring method is unsuitable due to the generation of undesired secondary radiation. While proton radiography is primarily used to probe transient electro-magnetic field structures, by definition it results in a structured proton beam. However, those experimental arrangements are generally complex as they often include at least two laser beams as well as two interaction targets, one to provide the proton probe and the other to generate sufficiently high electric fields in a high density plasma[34,35].

In this article, we report on an all-optical concept to modulate the profile of a multi-MeV proton beam with a single laser pulse by imprinting spatial intensity modulations of the laser onto the proton bunch, without significantly compromising the overall acceleration performance. Field maps induced by the TNSA drive laser itself in the residual gas of the interaction chamber are inscribed on the TNSA protons, as they probe these fields in a proton-radiography-like manner. The concept was motivated by an effect initially observed in an experiment dedicated to laser-driven proton acceleration from a renewable micrometer sized cryogenic Hydrogen jet target[36,37] at the Draco 150 TW laser[22]. In the experiment, prominent features of the collimated drive laser beam, such as the shadow of obstacles inserted deliberately in the beam, were observed to clearly reappear in the accelerated proton beam profile (Fig. 1). This observation is highly counter-intuitive for the following two reasons. First, such laser near-field intensity features do not express in the far-field, that is, the focus on the opaque target where hot electrons are generated. Including electron transport through the solid target, there is neither reason nor evidence for the formation of a correspondingly modulated sheath field. Simultaneous detection of the intensity distribution of the remnant laser light behind the target, however, showed identical yet edge-contrast enhanced features. This remnant laser light will be called transmitted light in the following for simplicity although the target remains opaque during laser interaction (transparency[38] is only expected for Hydrogen targets thinner than 250 nm at the given laser peak intensity). Data from the transmitted light diagnostic (ref. to Methods section) reveals that roughly 70% of the laser light propagated around the 5 μm diameter cylindrical Hydrogen jet. Blocking only the central area of a typical laser focus[39,40], indicated schematically in the inset in Fig. 1 with a high dynamic range Draco focus image, thus acts as an inverted Fourier-plane spatial filter, emphasizing high spatial frequency features as observed. This was confirmed in field propagation simulations that apply scalar diffraction theory to simulate the propagation of the laser field through space, in this case around obstacles in the collimated beam and the inverted filter in the laser focus (see Zemax calculation displayed in Fig. 1). Recent studies showed direct effects of the reflected or transmitted laser light co-propagating with relativistic electrons that originated in laser-solid interactions[41,42]. However, laser-accelerated proton bunches travel at velocities that are a fraction of the speed of light, which inherently precludes temporal overlap with the transmitted drive laser pulse when propagating away from the target. This breach of causality represents the second obstacle for an explanation of the measured proton features based on sheath field properties.

The imprinting of laser intensity features in the proton beam profile at macroscopic distances from the focus region, despite the lack of temporal overlap, can be explained by laser-induced quasi-static electric fields in the background gas of the experimental interaction chamber. Induced by ionization of residual gas molecules through the transmitted laser pulse up to >10 mm distances from the laser focus, these fields remain intact for tens of picoseconds, thereby serving as a memorizing structure to bridge the temporal gap until the arrival of protons that were accelerated at the target by TNSA-fields originating from the very same laser pulse. This basic concept is detailed in the following section. A dedicated experiment and particle-in-cell (PIC) simulations are presented to validate the proposed scheme.

## Results

**Imprint control by tuning the residual gas density around the target**. The scheme explaining the observed imprint effect is schematically depicted in Fig. 2a. Along the propagation axis of the transmitted laser pulse and the trailing proton bunch, three relevant regions are identified: region I contains the ultra-high intensity laser-target interaction, in this case TNSA, resulting in an accelerated proton bunch. Region II begins well beyond the focus and target region and reaches until the laser intensity has dropped, due to the beam divergence, below the ionization threshold of the residual gas molecules. The latter marks the transition to region III, in which no ionization takes place, hence is considered field-free.

Spatial filtering of the laser pulse by the opaque target in the laser focus results in strong laser transmission in high spatial frequency areas of the beam profile, whereas low spatial frequencies are suppressed. As a result, residual gas molecules across the transmitted laser beam are locally ionized in areas exhibiting high spatial laser frequencies, most prominently those

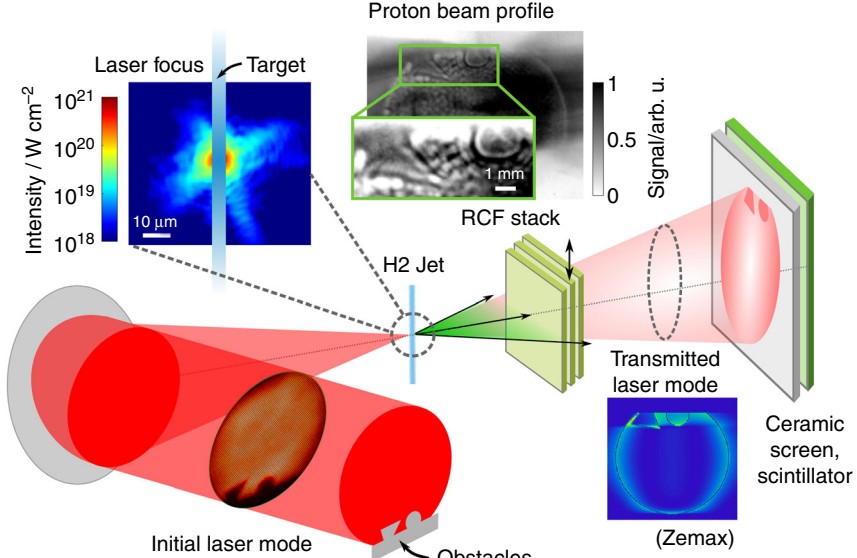

**Fig. 1** Schematic of experimental setup and example measurement. Obstacles were inserted in the collimated Draco laser beam before focusing it onto a micrometer sized solid Hydrogen jet target with an f/2.5 off-axis parabolic mirror. An image of the laser focus (logarithmic color scale) is overlapped with a schematic image of the Hydrogen jet (5 μm diameter) to visualize the amount of laser intensity present in the outer lobes of the focus. Proton beam profile measurements via radiochromic film (RCF) stacks inserted on-demand at 45 mm behind the target (7 MeV layer displayed, zoom-in on obstacles with different gray scale to emphasize imprinted structures) and a scintillator at 12.5 cm (exemplary data shown in Methods section) clearly reproduce the shape of the inserted obstacles, namely pick-off mirror and triangle. Zemax simulations confirm the effect of spatial filtering in the laser focus on the transmitted light intensity distribution that was measured by imaging a ceramic screen situated in front of the scintillator detector

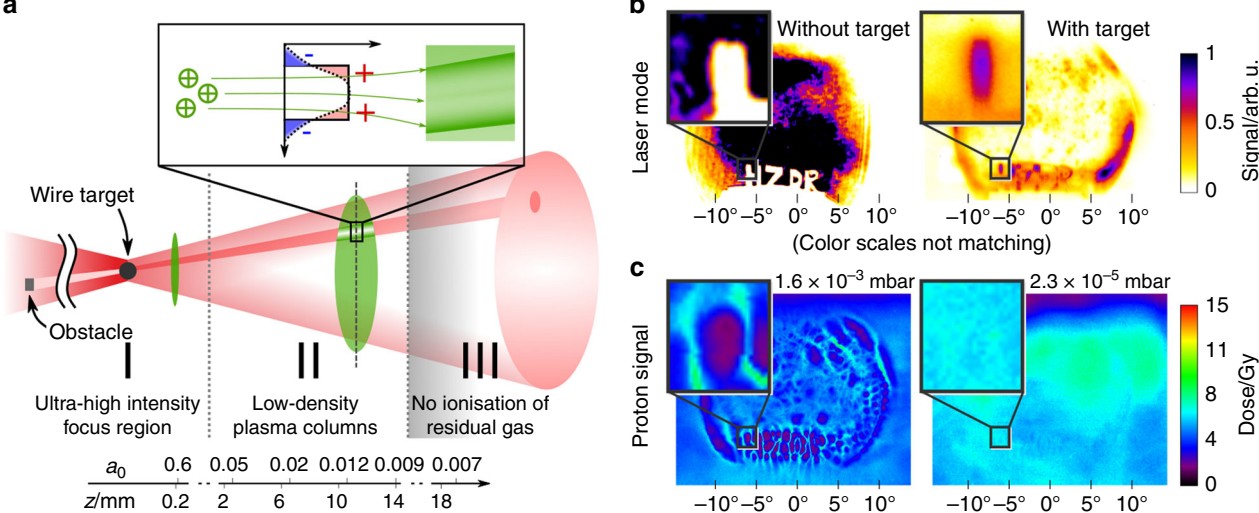

**Fig. 2** Imprint scheme and experimental imprinting results. **a** Schematic depiction of the imprint scenario, visualizing the transmitted laser path (red cone) and the accelerated proton bunch (green) along the propagation axis $z$. Between target and proton beam diagnostic, three main regions are identified and related to the transmitted laser intensity expressed via $a_0$. Spatial filtering in the laser focus results in the emphasis of high spatial laser frequencies, for example those related to obstacles inserted in the beam before focusing. In areas of sufficient transmitted laser intensity, residual gas molecules are ionized locally and low-density plasma columns extending along $z$ are formed until the laser intensity has dropped below the ionization threshold. In the quasi-neutral region II, transverse quasi-static electric fields between the plasma electrons and the remaining fixed ions are visualized via radiography with the accelerated protons as intrinsic probe. **b** Transmitted laser intensity mode without (left) and with (right) a 10 μm diameter tungsten wire positioned in the laser focus. The color scale for each picture was adjusted to ensure visibility of important features. **c** Proton beam profile (4.7 MeV layer displayed) at two different residual gas pressures in the experimental chamber. In the left image, structures corresponding to the obstacles inserted in the laser beam can be observed, as well as features originating from fluctuations intrinsic to the laser beam profile. By decreasing the pressure the imprint effect can be switched off almost completely (right image). The change in overall dose between both cases is within the range of shot-to-shot variations of the proton acceleration performance

inherent to diffraction of the laser at sharp edges of inserted obstacles in Fig. 1. Note that the ionization rate of residual gas molecules by hot electrons escaping the target is only in the order of one event $ms^{-1}$ $cm^{-3}$ and can therefore be omitted[43]. As the

transmitted laser pulse propagates through region II, which can reach up to >10 mm, depending on the given experimental parameters, these ionized areas extend to low-density plasma columns containing free electrons behind the laser pulse. The

plasma electrons gain energy in the field of the transmitted laser pulse[20,44], whereas the ions can be considered predominantly fixed, given the laser intensity is sufficiently low[45,46], which is the case for the experiments presented in this work. Charge displacement of free electrons with respect to the fixed ions[47] perpendicular to the laser and proton propagation axis, results in an electric potential that satisfies Poisson's equation $\Delta\Phi(r) \sim [n_e(r) - Z_i n_i(r)]$, where $Z_i n_i(r)$ and $n_e(r)$ are the respective one-dimensional ion and electron charge distributions. The corresponding electric fields $E_{trans}(r)$ are indeed quasi-static, as their persistence is only limited by the quasi-neutral plasma expansion[48] and the energy loss of the plasma electrons through electron-ion-collisions.

Both become relevant on a timescale of ~10 ns, leaving sufficient time for TNSA protons generated at the target to probe $E_{trans}$, for example, a 7 MeV proton arrives at $z = 2$ mm after ~55 ps. In the framework of laser-proton acceleration it can be assumed that the proton and laser beam divergence in the ionized areas in region II are sufficiently similar such that a proton experiences the deflecting fields of one low-density plasma column over the length of region II, leading to the observed sharp features in the final proton beam profile.

For the conditions of the Hydrogen jet experiment where the residual gas was dominated by Hydrogen vapor, the upper boundary of region II can be derived to be ~15 mm up to which ionization of the residual gas molecules takes place, corresponding to the appearance intensity $I_{app} = 1.37 \times 10^{14}$ W cm$^{-2}$. The lower boundary marking the transition to region I, while subject to ongoing investigations, is estimated heuristically: acknowledging that characteristic structures in the laser near-field intensity mode reappear in the recorded proton beam profiles, the imprinting effect has to occur at distances from the laser focus, clearly beyond the Fresnel region, where the laser profile has regained such features. As verified during the experiment, this only happens at a distance of roughly 500 μm behind the focus, resulting in an approximate length of region II of 14–15 mm (refer to Supplementary Notes 1 and 2 and Supplementary Figures 1 and 2 for further discussion). Along region II, the laser intensity decreases according to the beam divergence, which, given the experimental conditions, amounts to roughly one order of magnitude in terms of the laser field's dimensionless vector potential, $a_0$ (ref. to $a_0$ scale in Fig. 2a). A correspondingly moderate decay in field strength $E_{trans}$ with increasing distance from the laser focus along the low-density plasma columns is expected.

A first estimate for the field strength $E_{trans}$ is derived from the experimentally recorded proton beam profiles in the form of an example calculation. A proton with energy $\varepsilon_p$ that propagates through the electric field $\bar{E}_{trans}$ in region II with the field length $L$ is deflected by the amount $r_{def} = \frac{e}{4} \bar{E}_{trans} L^2 \varepsilon_p^{-1}$, followed by ballistic propagation in region III until the detector. Proton beam distortions of few 100 μm, as observed during the Hydrogen jet experiment, require mean deflecting fields of $10^6$–$10^7$ V m$^{-1}$ all along region II.

A dedicated experiment was conducted to validate the proposed scheme by unambiguously demonstrating that said electro-static fields are induced by ionization of the background gas in the experimental chamber, resulting in the transfer of the laser intensity profile information to the proton beam. Instead of the Hydrogen jet, causing an uncontrollable amount of Hydrogen vapor around the target, tungsten wire targets of 10 μm thickness were irradiated while actively tuning the background gas density (mainly Nitrogen), and hence plasma electron density $n_e$, with a precision air valve. For a chamber pressure of $1.6 \times 10^{-3}$ mbar, engineered laser beam profile modulations, in this case the letters "HZDR", reoccurred as edge contrast enhanced imprints in

the proton beam profile at the detector (ref. to Fig. 2b, c). Although the overall amount of transmitted light is reduced by roughly a factor two, resulting from the use of a wider target, remarkable agreement with the Hydrogen jet experiment is observed in the strong spatial filtering by the wire target in the laser focal plane. This expresses in the enhancement of the transmitted intensity of high spatial frequency contributions in the vicinity of the inserted letters, thereby defining the spatial intensity pattern for the ionization of the residual gas (Fig. 2b). As a result, areas of strong laser light transmission correspond directly to areas of low proton dose across the proton beam profile, as is displayed in the left image of Fig. 2c. Reduction of the vacuum chamber pressure to $2.3 \times 10^{-5}$ mbar resulted in nearly smooth proton beam profiles exhibiting no distinct signs of imprinting. This verifies that the quasi-static electric field strengths $E_{trans}$ scale with the electron density $n_e$ as was proposed above. The results of the tungsten wire experiment thereby confirm the important role of the residual background gas as a programmable medium for the laser intensity profile.

**2D-PIC simulations of $E_{trans}$ in low-density plasma**. Two-dimensional PIC simulations were performed to investigate the emergence and evolution of electro-static fields over the course of tens of picoseconds after the propagation of a laser pulse through residual gas in region II. This allowed for the quantitative study of maximum field strengths and lateral distributions of $E_{trans}$, both expected to depend strongly on the local electron density distribution. Two different gas settings utilizing Hydrogen gas with densities roughly corresponding to the vacuum chamber pressures realized during the tungsten wire experiment were studied: $n_e = 10^{14}$ cm$^{-3}$ and $n_e = 10^{12}$ cm$^{-3}$, corresponding to chamber pressures $1.6 \times 10^{-3}$ mbar and $1.6 \times 10^{-5}$ mbar, respectively (assuming three to fourfold ionization of Nitrogen molecules, ref. to Supplementary Figure 1 for details). Intensity modulations on top of a plane wave laser mode were modeled with a sinusoidal function for simplicity, the period of which was adapted to roughly reproduce the characteristic size of the intensity modulation in the experiment, namely, ~1 mm at the position of the proton beam diagnostic and scaled along the laser axis according to the laser beam divergence. Several simulations varying $a_0$ and the feature size, thereby representing different distances from the focus were performed. As the decay in laser intensity according to experimental conditions was relatively slow within one simulation window, no laser divergence angle was included for the separate simulations.

Transverse electric field maps of a single plasma column at two different time steps are displayed in Fig. 3, illustrating the field evolution on a few 10 ps-scale for both plasma density cases. The filament diameter and maximum laser intensity correspond to estimated experimental conditions at a distance 2 mm behind the laser focus. Sharp transverse electric fields are formed at the border between the ion population remaining immobile and the free electron plasma, with electron temperatures in the order of few hundred eV, that is distributed over a larger lateral area. Maximum electric field strengths are few times $10^6$ V m$^{-1}$ for $n_e = 10^{14}$ cm$^{-3}$ and roughly a factor 40 lower for $n_e = 10^{12}$ cm$^{-3}$, agreeing well with estimated values in the previous section based on experimentally observed proton beam deflections. The orientation of the transverse electric fields are such that protons propagating along region II would be deflected radially outwards of the columns, resulting in low proton signal in areas of high transmitted light intensity, in agreement with experimental results (Fig. 2b, c).

Within the boundaries of region II that were estimated in the previous section, maximum field strengths are nearly constant

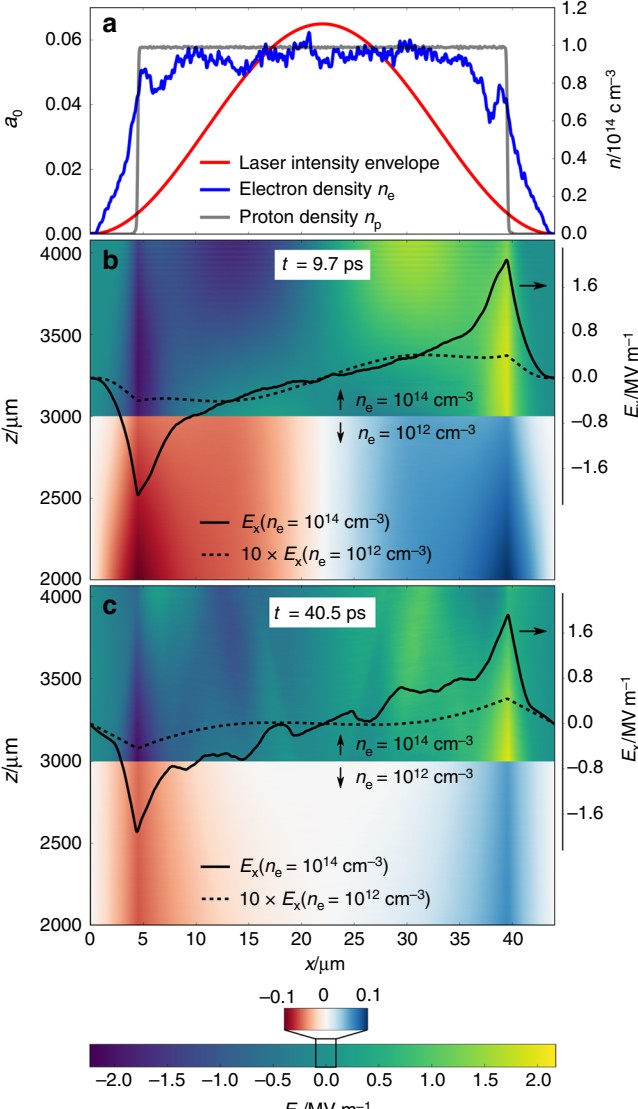

**Fig. 3** 2D-PIC simulation results of transverse electric field maps in a single plasma column. The field distribution develops as an intensity modulated plane wave laser pulse travels along $z$ through Hydrogen gas of different densities. The peak intensity of $a_0 = 0.065$ resembles experimental laser conditions at a distance $z = 2$ mm after the laser focus. The laser pulse is initialized at the bottom of the simulation window at time $t = 0$. **a** Laser intensity envelope (red) with a lateral size of 44 μm and line-outs of both electron (blue) and proton densities (gray) for the electron density case of $n_e = 10^{14}$ cm$^{-3}$ at time step 9.7 ps. Step-like ionization of Hydrogen atoms occurs at the appearance intensity, that is, $a_0 = 0.008$. Note that, while the proton density gradient appears to be rather sharp on the few 10–100 μm scale relevant to this study, it is in fact smooth when zooming in on μm distances, as is expected from a probabilistic ionization model such as ADK (ref. to Methods section). **b** and **c** Electric field distribution in $x$-direction at time steps 9.7 ps and 40.5 ps, respectively, for electron densities $n_e = 10^{14}$ cm$^{-3}$ (top) and $n_e = 10^{12}$ cm$^{-3}$ (bottom). Line-outs of the respective field distribution at $z = 3220$ μm are overlayed as black solid and dashed curves, where the latter is scaled by a factor 10

along $z$ (see Supplementary Note 2 and Supplementary Figure 2 for details). Until the final time step at ~35 ps after the laser has left the simulation window, no decrease in field strength is observed. Therefore, it can be concluded that the fields remain present for considerably longer timescales, allowing for them to be probed by protons accelerated at the target.

## Discussion

We presented an all-optical concept for the effective beam profile manipulation of laser-accelerated proton bunches, relying on asynchronous information transfer from the spatial profile of the high power drive laser pulse to the proton bunch. Found to be independent from the original acceleration process, two essential ingredients of this concept could be demonstrated. Numerical simulations confirmed the hypothesis of the formation of drive laser-induced persistent field structures in the background gas downstream of the laser target interaction, capable of locally deflecting protons. Experimental variation of the gas pressure proved to control the proton beam structure in pressure ranges typically occurring in laser plasma experiments. The intense light required for the structured downstream ionization of the background gas was identified to be leaking around finite sized targets. As such, the target represents a spatial frequency filter, very similar to the well-known Schlieren-imaging technique, that enhances the contrast of obstacles intentionally placed in the original laser beam.

Apart from deploying laterally limited targets, other scenarios can be considered where laser light reaches the proton propagation region downstream of the ultra-high intensity laser-target interaction. Naturally, this is the case for transparent targets, e.g. foils that are thinner than the skin depth of the evanescent laser wave in the plasma, or even for thicker targets due to the relativistic mass increase of the plasma electrons[23,38,49,50], often referred to as relativistic-induced transparency (RIT). There, the so far discussed inverted spatial filter function in the Fourier plane is inverted back to normal as the target turns transparent initially in the center[42,51], leading to a different transmitted light mode than when deploying opaque but laterally small targets[33]. Through careful shaping of its spatio-temporal density distribution, the target can be used as a tunable spatial frequency filter.

The amount of transmitted light, residual gas density and constituents, as well as the particular shape of the incoming laser intensity mode and its filtering via the tunable spatial frequency filter, represent straightforward, effective, and mostly independent tuning knobs to control the imprinted modulations in the proton beam. Strong masking of the incoming laser light will ultimately limit the proton acceleration performance, therefore, clever mask geometries will be needed for efficient proton beam structuring. Notably, nonlinear plasma processes like filamentation instabilities, potentially occurring during the acceleration phase (region I in Fig. 2), may inherently modulate the proton beam profile and thereby superimpose or even overwrite intentional imprint structures, engineered in region II. Examples include beam modulations observed to be generated in the ultra-high contrast laser-driven radiation-pressure dominated regime[52,53], but also in the case of longer pre-plasma scale lengths[6,54,55]. According to our results, distinguishing these effects from all-optical imprinting by the same laser driving the proton acceleration can be ambiguous when only proton beam profiles are evaluated. Conclusions that were derived from experimental beam profile modulations in past studies, detailing laser-plasma interaction parameters that lead to filamentation instabilities, may need to be revisited. However, simultaneous observation of the transmitted laser light and assessment of the influence of different residual gas concentrations will facilitate future interpretation. Whereas under the presence of such instabilities proton beam quality control is challenging, programmable laser imprinting is easily controllable under standard experimental conditions.

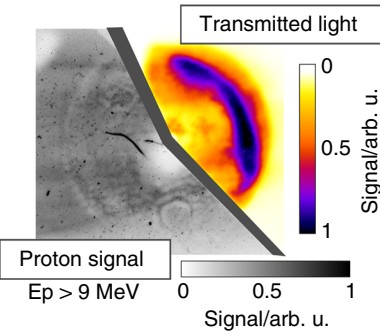

**Fig. 4** Example measurement of transmitted laser light and proton beam profile on the same shot. Protons with energies higher than 9 MeV are detected

Future studies will explore the accessible parameter space of the all-optical proton beam structuring mechanism, based on the framework laid out here. Potential applications that would benefit from structured proton beams lie in laser-driven proton radio-oncology[56,57], ion implantation[58,59], stress testing of materials[10], or isochoric heating in warm dense matter research[11]. Further systematic studies that are concerned with shaping proton beam profiles to a customized target design are required to harness this beam structuring approach for applications.

## Methods

**Laser and target parameters**. The experiment was performed with the Titanium: Sapphire-based laser system Draco[22] providing 30 fs long pulses at ~3 J laser energy after final pulse compression. A single plasma mirror setup resulted in a temporal contrast improvement of two to three orders of magnitude, resulting in intensity values normalized to the peak intensity of $\leq 10^{-12}$ at $t = -120$ ps and $\sim 10^{-7}$ at $t = -2$ ps before the arrival of the main laser pulse (at $t = 0$), respectively. The target surface undergoes ionization at $t \approx -3$ ps, so the peak laser intensity interacts with a largely unexpanded target[60]. Near-field measurements of the laser intensity mode behind the plasma mirror were recorded on each shot and did not reveal modulations that would have resulted in a reduction of final focus quality.

Obstacles were inserted in the collimated laser beam between the plasma mirror setup and the final focusing optic (ref. to Fig. 1). The laser beam was focused onto the target by a f/2.5 off-axis parabolic mirror to a spot size of 3 μm full width half maximum, measured over roughly two orders of magnitude in intensity with a 12 bit CCD camera. This results in a peak intensity on target of approximately $6 \times 10^{20}$ W cm$^{-2}$ and a normalized vector potential of $a_0 \sim 16$. Two different targets were used in this study: a solid Hydrogen jet[61] of cylindrical footprint with a diameter of 5 μm and tungsten wires of 10 μm diameter. The Hydrogen jet is produced by injecting cryogenically cooled liquid Hydrogen through a micrometer sized aperture into vacuum where it solidifies due to evaporative cooling. Flow velocities in the order of 100 m s$^{-1}$ allow for high laser shot rates[37].

**Diagnostics**. Radio-chromic film (RCF) stacks to diagnose the proton beam profile on a single shot basis were inserted on-demand along the laser propagation direction at a distance of 45 mm behind the target. In case no RCF-stack was irradiated, the transmitted laser light was recorded by means of a ceramic screen installed at a distance of 125 mm behind the target and imaged onto a CCD camera equipped with an interference filter optimized to the Ti:Sa wavelength spectrum and located outside the experimental chamber.

Stacked behind the ceramic screen was a scintillator, able to detect protons with energies ≥9 MeV. This combined measurement allows for an on-shot correlation between the laser-accelerated proton beam profile and the transmitted laser mode, refer to Fig. 4. A clear reduction in proton signal in regions of high laser light transmission is observed. The proton beam diagnostics were complemented by a Thomson Parabola energy spectrometer aligned along the laser axis and equipped with micro channel plates to enable fast readout on a multi-shot basis.

**Simulation**. Two-dimensional particle-in-cell (PIC) simulations were carried out with PIConGPU[62] version 0.3.0 for two different background densities: $n_e = 10^{12}$ cm$^{-3}$ and $n_e = 10^{14}$ cm$^{-3}$. The spatial resolution was 23.3 cells (34.375 nm) per laser wavelength (800 nm) in all dimensions ($\Delta t = 81$ as). Particle-field interaction was modeled with TSC shape and two weighted, initially neutral ions (Hydrogen) per cell. Ionization of the residual gas was included according to a probabilistic ADK model[63] with BSI cutoff[64]. The laser pulse was modeled as an analytical background field, polarized in the plane of the simulation with a pulse length of 30 fs. In the transverse direction, a sinusoidal intensity modulation was applied to model intensity

variations across the laser near-field. The dimensionless laser vector potential $a_0$ was varied accounting for the laser beam divergence when propagating away from the focus region. Periodic boundary conditions were applied for the transverse direction. PIConGPU is open source and all versions, including 0.3.0, are available as download and contributable git repository[65].

## Data availability
Experimental and simulation data that support the findings of this study, as well as simulation input files are available on RODARE with the identifier https://doi.org/10.14278/rodare.66[66]

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

## Acknowledgements

S. Bock, R. Gebhardt, U. Helbig, and T. Püschel are highly acknowledged for their excellent laser support. Furthermore, we would like to acknowledge all contributors to the open-source code PIConGPU for enabling our simulations. Computations were performed on the Hypnos cluster at the Helmholtz-Zentrum Dresden - Rossendorf. We thank A. Debus for fruitful discussions. Furthermore, we acknowledge T. Laarmann (DESY Hamburg) for providing the cryostat of the Hydrogen jet system, as well as J. Tiggesbäumker, L. Kazak, and S. Wolter for technical support during the Hydrogen jet campaign. This project has received funding from the European Union Horizon 2020 research and innovation programme (contract No. 654220) and LASERLAB-EUROPE/LEPP (contract No. 654148), as well as the U.S. Department of Energy Office of Science, Fusion Energy Sciences (FWP 100182, FWP 100237, and FWP 100331) and the Sonderforschungsbereich 652 of the German Science Foundation (DFG). C.R. acknowledges financial support from the Volkswagen Foundation. C.B.C. acknowledges partial support by the Natural Sciences and Engineering Research Council of Canada (NSERC).

## Author contributions

L.O.-H., T.Z., F.-E.B., C.B.C., M.Gauthier, S.G., A.I., J.B.K, S.K., F.K., J.M.-N., I.P., M.R., C.R., H.-P.S., and K.Z. conducted the experiments. T.K., J.B., R.P., and A.H. performed numerical PIC simulations. M.Garten implemented ionization models in PIConGPU. T. Z. carried out Zemax simulations. L.O., T.Z., T.E.C., T.K., A.H., C.R., and K.Z. interpreted the experimental and numerical results and performed analytical calculations. F.F. provided support in the evaluation of the results. M.B., T.E.C., S.H.G., U.S., and K.Z. supervised the project. All authors contributed to discussions and revision of the manuscript.

## Additional information

**Competing interests:** The authors declare no competing interests.

