## [Peer Review File · Nature Communications]

Reviewers' comments:

Reviewer #1 (Remarks to the Author):

The manuscript by Obst et al reports on an interesting novel approach for structuring laser-accelerated proton beams by optical imprint of electric field structure in the background gas of the evacuated interaction chamber. The structures are imprinted, in the arrangement used, by laser light transmitted at the sides of the target used to accelerated the protons (which is either a 5 μ m diameter cryogenic jet, or a 10 μ m metallic wire).

The idea is original and elegant, and its capabilities are demonstrated by the authors through the deliberate imprinting of a logo into the proton beam. I found this paper very interesting and , while this implementation employs a single laser beam, even broader capabilities could be introduced by using a second, low intensity laser pulse to condition the propagation of the protons - a discussion of this option could perhaps be added to the manuscript.

I found the manuscript very interesting, well written and rigorous in data presentation, analysis and discussion. Excellent 2D PIC model simulations are provided in support of the interpretation.

I congratulate the authors for this excellent research work, and support publication of this manuscript, which I trust will be of interest to the readership of Nature Communications.

Two comments/queries:

1) I assume that, particularly when using the 10 μ m wire, most of the laser energy is stopped by the target and only a small fraction (associated to the focal spot outer wings) makes it to region II. How is the laser intensity calculated in this region? The intensity decrease due to beam divergence is discussed, but could the author discuss how the attenuation caused by the target is taken into account?

2) When discussing the modulations imprinted on the proton beams by laser-transmission through a transparent target at page 6, it may be appropriate to quote the following paper: B. Gonzalez-Izquierdo, et al, Towards optical polarization control of laser-driven proton acceleration in foils undergoing relativistic transparency, Nature Comm., 7, 12891 (2016), which, following from ref [39], focuses on structures imprinted on the proton beam profile during relativist transparency

Reviewer #2 (Remarks to the Author):

The manuscript by L. Obst et al presents an all-optical approach for shaping the spatial profile of a laser-driven ion beam. The ion acceleration occurs in the TNSA regime. Compared to typical TNSA experiments, an amplitude mask is used to modulate the intensity profile of the laser pulse that drives the proton accelerator. The laser is focused on a wire target from which protons are accelerated. The wire diameter being smaller than the laser waist, an important part of the laser energy pass trough the target which acts as a high-pass spatial-filter for the laser. This leads to a magnification of the regions of high intensity gradients. This intensity profile imprints on the plasma density profile of the residual plasma behind the target (after plasma expansion). Eventually, the proton beam is deflected as it goes through this modulated plasma.

The technique is new to my knowledge and the results are impressive. In particular, the fact that the intensity modulations in the laser profile can be imprinted to the proton beam is unexpected and

striking. Yet, I do not see clearly how this approach may have practical applications and I would like the authors to elaborate more on this point. They may especially describe a mask that would lead to an efficient steering. Some critical points should also be clarified (notably the point 3, on the validity of the simulations):

1. The first question regards the energy efficiency.

1.a. In Fig. 2c, the integrated charge seems lower in the case with the mask. More generally, a significant structuring of the proton beam would imply a large waste of laser-energy (large part has to be blocked).

1.b. A significant part of the energy should pass the target for an efficient steering.

For both reasons, a large part of the energy is not available for the acceleration, which questions the potential of the scheme.

2. The method magnifies the intensity gradients in the laser beam profile. As a result inhomogeneities in the laser energy distribution (without mask) lead to structures in the proton beam. This is a second important shortcoming of the approach: laser defects are magnified and imprinted in the proton beam.

3. In simulations, H₂ is used instead of N₂ in the experiment. The consequences of this choice should be more discussed. With N₂, I would expect the ionization level, and hence the electron density to depend more strongly on the laser intensity, and hence to vary with z . Anyway and for any gas, the strong variations of the laser intensity (about 5 order of magnitude over 10 mm), should lead to large variations of the density and/or the temperature. The fact that the plasma field remains almost constant over ~ 4 mm in Suppl. Fig. 1b is thus unexpected and should be explained.

Maybe I misunderstood this part, but it seems also to me that a few of hundred of eV temperature is not consistent with three to four-fold ionization of Nitrogen molecules (electrons with energy > 98 eV would lead to five-fold ionization).

4. In the 'wire' experiment the wire diameter (10 μ m) is large compared to the laser beam (3 μ m FWHM). The percentage of the laser energy going around the target should therefore be much smaller than in the 'hydrogen jet' experiment. This may be commented. The fact that 70% of the energy passes around the target in the case of the hydrogen target is also a bit weird. For a FWHM of 3 μ m, 86% of the laser energy is expected to be in a diameter of 5,1 μ m.

5. More details should be given on the Zemax simulation. Page 2, ray-tracing simulations are mentioned. Such simulations does not seem appropriate for Fourier filtering.

6. The control of the beam divergence and direction may be challenging with existing techniques but they are not clearly demonstrated in the paper. More caution in the introduction would be appreciate. Further, the term 'model' in the first part seems a bit excessive. I would rather use 'scenario', 'analysis, or 'interpretation'...

Reviewer #3 (Remarks to the Author):

The manuscript "All-optical structuring of laser-driven proton profiles" by L. Obst et al. reports on proton radiography of laser-induced electric fields in an underdense plasma. Proton radiography is a well-established technique in the field, especially to measure electric or magnetic field structures generated by intense lasers in plasmas (e.g. Borghesi PoP 9, 2214 (2002), PRL 92, 055003 (2004),

[6,7]) and more relevant for this manuscript, to visualize the charge-displacement channel formation dynamics following relativistic self-focusing of laser pulses (Kar NJP 9 402 (2007), Willingale PRL 106, 105002 (2011)). In the present manuscript, such a pump probe setup was realized with a single focused laser pulse. The central part of the laser pulse in the focal plane was used to accelerate the proton beam from a thin cylindrical H₂ jet/ wire target. The outer parts of the laser (missing the wire) ionized background gas downstream of the target which induced electric fields that in turn are probed by the slower proton beam. Objects are then inserted in the outer region of the near field of the laser which visualize in the proton profiles, because the protons can propagate undisturbed in the absence of ionizing laser light in the low density gas. Such an experiment configuration is very common (see above references) with the neat exception that the authors realize this with a single laser pulse. The electric fields probed are also not exceptional in structure or magnitude.

In view of the above, novelty and significance of this work are limited and only represent an incremental advancement of the field. The authors' argumentation of "direct control of the proton profile" or "novel all-optical concept to modulate the profile of the [...] proton beam" is misleading and they appear to be overselling their claims. In conclusion, I cannot recommend publication in Nature Communication.

Answers to the Reviewers

Reviewer #1

Reviewer: The manuscript by Obst et al reports on an interesting novel approach for structuring laser-accelerated proton beams by optical imprint of electric field structure in the background gas of the evacuated interaction chamber. The structures are imprinted, in the arrangement used, by laser light transmitted at the sides of the target used to accelerate the protons (which is either a 5 μ m diameter cryogenic jet, or a 10 μ m metallic wire).

The idea is original and elegant, and its capabilities are demonstrated by the authors through the deliberate imprinting of a logo into the proton beam. I found this paper very interesting and, while this implementation employs a single laser beam, even broader capabilities could be introduced by using a second, low intensity laser pulse to condition the propagation of the protons - a discussion of this option could perhaps be added to the manuscript.

I found the manuscript very interesting, well written and rigorous in data presentation, analysis and discussion. Excellent 2D PIC model simulations are provided in support of the interpretation.

I congratulate the authors for this excellent research work, and support publication of this manuscript, which I trust will be of interest to the readership of Nature Communications.

Answer: We would like to express our thanks to Reviewer #1 for his/her appreciation of our work and recommendation for publication in Nature Communications. Indeed an important appeal of the reported laser-imprinting effect lies in its occurrence when deploying only a single laser pulse. However, the same effect could in principle be achieved with a secondary laser without significant changes to the result, as long as the proton bunch and secondary laser beam are aligned over a sufficiently long "co-propagation" axis (region II in our manuscript) for effective imprinting to occur. Naturally, this then resembles a radiography experiment at lower gas density than is usually reported on.

R: Two comments/queries:

1) I assume that, particularly when using the 10 μ m wire, most of the laser energy is stopped by the target and only a small fraction (associated to the focal spot outer wings) makes it to region II. How is the laser intensity calculated in this region? The intensity decrease due to beam divergence is discussed, but could the author discuss how the attenuation caused by the target is taken into account?

A: In both experiments, deploying a 5 μ m diameter cryogenic Hydrogen jet in the first experiment and a 10 μ m diameter tungsten wire in the second, a transmitted light diagnostic was installed downstream (refer to Methods section of our manuscript for details). This diagnostic showed that when deploying the Hydrogen jet, roughly 70 % of the initial laser energy was transmitted. When using the tungsten wire this value is reduced by about a factor two, resulting in values between 30 % and 40 % transmission. This information was so far not given in our manuscript, so we changed the following sentence:

"In remarkable agreement with the Hydrogen jet experiment, the strong spatial filtering by the wire target in the laser focal plane enhances the transmitted intensity of high spatial frequency contributions in the vicinity of the inserted letters, and thereby defines the spatial intensity pattern for the ionization of the residual gas (see Fig. 2b)."

to

“While the overall amount of transmitted light is reduced by roughly a factor two, resulting from the use of a wider target, remarkable agreement with the Hydrogen jet experiment is observed in the strong spatial filtering by the wire target in the laser focal plane. This expresses in the enhancement of the transmitted intensity of high spatial frequency contributions in the vicinity of the inserted letters, thereby defining the spatial intensity pattern for the ionization of the residual gas (see Fig. 2b).” (page 4 lines 142 ff. of the revised manuscript¹)

R: 2) When discussing the modulations imprinted on the proton beams by laser-transmission through a transparent target at page 6, it may be appropriate to quote the following paper: B. Gonzalez-Izquierdo, et al, Towards optical polarization control of laser-driven proton acceleration in foils undergoing relativistic transparency, Nature Comm., 7, 12891 (2016), which, following from ref [39], focuses on structures imprinted on the proton beam profile during relativist transparency

A: We thank Reviewer #1 for the suggestion and added the reference to our manuscript.

Reviewer #2

Reviewer: The manuscript by L. Obst et al presents an all-optical approach for shaping the spatial profile of a laser-driven ion beam. The ion acceleration occurs in the TNSA regime. Compared to typical TNSA experiments, an amplitude mask is used to modulate the intensity profile of the laser pulse that drives the proton accelerator. The laser is focused on a wire target from which protons are accelerated. The wire diameter being smaller than the laser waist, an important part of the laser energy pass through the target which acts as a high-pass spatial-filter for the laser. This leads to a magnification of the regions of high intensity gradients. This intensity profile imprints on the plasma density profile of the residual plasma behind the target (after plasma expansion). Eventually, the proton beam is deflected as it goes through this modulated plasma.

The technique is new to my knowledge and the results are impressive. In particular, the fact that the intensity modulations in the laser profile can be imprinted to the proton beam is unexpected and striking.

Answer: We thank Reviewer #2 for his/her in-depth assessment of our work and for acknowledging the impact and novelty of our results, as well as his/her comments that helped us improve and clarify our manuscript.

R: Yet, I do not see clearly how this approach may have practical applications and I would like the authors to elaborate more on this point.

A: In this manuscript we aim to present a proof-of-concept study to describe a so far unrecognized effect in laser-driven proton acceleration with small targets. However, we do not claim to present a “proto-type” setup dedicated to a certain application. The goal of this works is to promote a new technique to substructure laser-driven proton beams, not least to eventually receive inspiration from other communities, such as material or health science, concerning operation parameters that would be necessary to answer the needs of a certain application. Future studies could then be concerned with demonstrating effectiveness of this technique to a certain field of research.

¹ Throughout this answer letter, references to page and line numbers will apply to the document Marked_Up_Manuscript.pdf where we highlighted changes made to the originally submitted manuscript. For readability, we also submitted the Main_manuscript_rev.pdf as the revised manuscript, where changes are not marked anymore.

Nevertheless, we believe that potential applications lie in radio-oncology², ion implantation³, stress testing of materials⁴, or for isochoric heating in warm dense matter research⁵. Specifically in tumor treatment, localized proton irradiation could be realized with all-optical proton beam structuring to spare healthy tissue, while inserting metal masks in the proton beam would result in undesired secondary radiation. We made the following changes to our manuscript to incorporate these potential applications:

Introduction

- *“Influencing the symmetry of the accelerating electron sheath, demonstrated by shaping of the focal spot, by introducing a laser pulse front tilt, or by micro-engineering of the target surface, while key to control, is often difficult to realize in application-oriented experiments.”* (for readability references from the manuscript are not included in this answer letter)
was changed to
“Influencing the symmetry of the accelerating electron sheath has been demonstrated by shaping of the focal spot, by introducing a laser pulse front tilt, or by micro-engineering of the target surface.” (page 1 lines 41 ff. of revised manuscript⁶)
- we added: *“These approaches, while key to control proton beam propagation, are often difficult to realize in application-oriented experiments, and, more importantly, do not allow tailoring of structures on the generated beam profile according to a specific design. Particularly, blocking dose in certain areas across the proton beam could be of significant interest for those applications where inserting a metal mask into the particle beam as an alternative beam structuring method is unsuitable due to the generation of undesired secondary radiation.”* (page 1 lines 47 ff. of the revised manuscript)

Discussion and outlook

- we added: *“Potential applications that would benefit from structured proton beams lie in laser-driven proton radio-oncology, ion implantation, stress testing of materials or isochoric heating in warm dense matter research. Further systematic studies that are concerned with shaping proton beam profiles to a customized target design are required to harness the novel beam structuring approach for applications.”* (page 7 lines 217 ff. of revised manuscript, refer to revised manuscript for references)

Another important application lies in the interpretation of modulated proton beam profiles generated in past and future laser-proton acceleration experiments, as we describe briefly on page 6 lines 204 ff. Though not mentioned extensively in the manuscript that is primarily aimed at investigating and establishing the mechanism itself, all-optical laser imprinting by the laser that drives the proton acceleration can superimpose proton beam modulations arising from plasma instabilities at the target. The latter are studied extensively, for example in the context of radiation pressure acceleration with extremely thin foils, as they potentially limit the application of laser-driven protons as well as predictable scalings of such novel accelerators (refer to manuscript for references). As all-optical laser imprinting in the residual vacuum chamber gas occurs under conditions that are very common in laser-plasma experiments, some studies that are concerned with filamented proton beams may even require revisiting. We demonstrate that ultimately, separating all-optical laser imprinting from plasma instability-related proton beam modulations (and, hence, proper interpretation of the latter) requires

² **Ledingham, K. W. D. & Galster, W.** Laser-driven particle and photon beams and some applications. *New Journal of Physics* 12, doi:10.1088/1367-2630/12/4/045005 (2010); **Kraft, S. D. et al.** Dose-dependent biological damage of tumour cells by laser-accelerated proton beams, *New Journal of Physics* 12, doi: 10.1088/1367-2630/12/8/085003 (2010); **Malka V. et al.** Practicability of proton therapy using compact laser systems, *Phys. Rev. Lett.* 31(6), doi: 10.1118/1.1747751 (2004)

³ **Boody, F. P. et al.** Laser-driven ion source for reduced-cost implantation of metal ions for strong reduction of dry friction and increased durability, *Laser Part. Beams* 14 doi:10.1017/S0263034600010132 (1996); **Torrioni, L. et al.** Implantation of ions produced by the use of high power iodine laser, *Appl. Surf. Sci.* 217, doi: 10.1016/S0169-4332(03)00551-8 (2003)

⁴ **Barberio, M. et al.** Laser-accelerated particle beams for stress testing of materials, *Nat. Commun.* 9 doi: 10.1038/s41467-017-02675-x (2018)

⁵ **Patel, P. et al.** Isochoric heating of solid-density matter with an ultrafast proton beam, *Phys. Rev. Lett.* 91, doi: 10.1103/PhysRevLett.91.125004 (2003)

⁶ Throughout this answer letter, references to page and line numbers will apply to the document `Marked_Up_Manuscript.pdf` where we highlighted changes made to the originally submitted manuscript. For readability, we also submitted the `Main_manuscript_rev.pdf` as the revised manuscript, where changes are not marked anymore.

controlling vacuum conditions and monitoring the transmitted laser light. We made the following change to the Discussion and outlook section to emphasize the impact on earlier studies concerned with proton beam profile modulations:

We changed "*Distinguishing these effects can be ambiguous when only proton beam profiles are evaluated, yet, simultaneous observation of the transmitted laser light and assessment of the influence of different residual gas concentrations will facilitate future interpretation.*"

to

"According to our results, distinguishing these effects from all-optical imprinting by the same laser driving the proton acceleration can be ambiguous when only proton beam profiles are evaluated. Conclusions that were derived from experimental beam profile modulations in past studies, detailing laser-plasma interaction parameters that lead to filamentation instabilities, may need to be revisited. However, simultaneous observation of the transmitted laser light and assessment of the influence of different residual gas concentrations will facilitate future interpretation." (page 7 lines 208 ff. of the revised manuscript)

R: They may especially describe a mask that would lead to an efficient steering.

A: We would like to point out that we do not claim *steering* but rather *structuring* of the proton beam, i. e. shifting of the proton dose in selected areas across the beam profile by $\leq 1^\circ$. Which masks would lead to satisfactory proton beam structures remains to be investigated and clearly depends on the required proton beam profile by a specific application as indicated above. As stated correctly by the reviewer in the next comment, too much blockage of the initial laser beam can lead to a decrease in proton acceleration performance, expressing in the reduction in particle numbers and energies. Careful scanning of different laser masks and spatial filter functions in the laser focus will be necessary to prepare laser-imprinting for a specific application. These studies are yet to be performed.

R: Some critical points should also be clarified (notably the point 3, on the validity of the simulations):

1. The first question regards the energy efficiency.

1.a. In Fig. 2c, the integrated charge seems lower in the case with the mask. More generally, a significant structuring of the proton beam would imply a large waste of laser-energy (large part has to be blocked).

A: Both pictures in Fig. 2c were taken with the mask inserted in the incoming laser beam, only the vacuum chamber pressure was varied with the precision air valve. The change in overall proton charge between the two shots displayed in Fig. 2c amounts to a maximum factor 1.5, which is well within shot-to-shot fluctuations of the proton acceleration. Nevertheless, as pointed out above, an optimum between maximum blockage of laser energy and efficient structuring of the proton beam will need to be investigated in future studies. However, we would like to draw attention to the fact that structuring occurs at the *edges* of objects inserted into the beam, due to the diffraction of laser light and the subsequent high-pass filtering of the diffraction pattern. Therefore, efficient structuring is not necessarily related to large blocked areas in the laser beam but rather to clever arrangements of edges for example in grid structures. Moreover, the spatial filtering in the focal plane can further be optimized by variation of the target size, thereby allowing for the selection of spatial frequencies which potentially relate to the size of the imprinted structures.

We made alterations in the following sections of our manuscript for clarification:

Imprint control by tuning the residual gas density around the target:

- we changed: "*As a result, residual gas molecules across the transmitted laser beam are locally ionized in areas exhibiting high spatial laser frequencies, most prominently those inherent to sharp edges of inserted obstacles in Figure 1.*"

to

"As a result, residual gas molecules across the transmitted laser beam are locally ionized in areas exhibiting high spatial laser frequencies, most prominently those

inherent to diffraction of the laser at sharp edges of inserted obstacles in Figure 1.” (p. 3 lines 100 ff. of the revised manuscript)

- we added the following sentence to the caption of Fig. 2c) of the revised manuscript: *“The change in overall dose between both cases is within the range of shot-to-shot variations of the proton acceleration performance.”*

Discussion and outlook

- we added: *“Strong masking of the incoming laser light will ultimately limit the proton acceleration performance, therefore, clever mask geometries will be needed for efficient proton beam structuring.”* (page 6 lines 202 ff. in the revised manuscript)

R: 1.b. A significant part of the energy should pass the target for an efficient steering. For both reasons, a large part of the energy is not available for the acceleration, which questions the potential of the scheme.

A: We agree with the reviewer that laser light passing by the target at a significant lateral distance does not contribute to the proton acceleration mechanism. However, it is generally understood that absorption of laser energy into the target plasma and the resulting proton beam performance is a highly complex mechanism depending on various factors such as laser intensity, laser contrast, target material and target geometry. In an earlier experiment, we demonstrated that comparable maximum proton energies (almost 20 MeV) were reached from 2 μm thick and laterally large titanium foils, 20 μm wide and 2 μm thick planar Hydrogen jets as well as 5 μm diameter cylindrical Hydrogen jets, all under optimized TNSA conditions⁷. A recent study even showed the benefits of a laser focus of approximately the same size as the target where laser light practically “engulfs” the target plasma, resulting in the acceleration of high-dose proton bunches of narrow energy spread⁸. Other studies have indicated that a grazing laser incidence geometry is favorable when aiming for maximum proton energies⁹. More generally, the laser-proton acceleration community has not yet identified the optimal laser-target configuration to generate highest proton energies and numbers in a reproducible manner. Therefore, we refrain from stating limitations to laser-imprinting, which relate proton beam performance to target size and shape.

To clarify that the acceleration performance was not affected by masking the incoming laser beam in our experiments we changed the following sentence in the revised manuscript:

- *“In this article we report on a novel all-optical concept to modulate the profile of a multi-MeV proton by imprinting spatial intensity modulations of the drive laser onto the proton bunch.”*
to
“In this article we report on a novel all-optical concept to modulate the profile of a multi-MeV proton beam with a single laser pulse by imprinting spatial intensity modulations of the laser onto the proton bunch without significantly compromising the overall acceleration performance.” (page 2 lines 56 ff. of the revised manuscript)

R: 2. The method magnifies the intensity gradients in the laser beam profile. As a result inhomogeneities in the laser energy distribution (without mask) lead to structures in the proton beam. This is a second important shortcoming of the approach: laser default are magnified and imprinted in the proton beam.

A: The reviewer correctly states that intensity modulations inherent to the laser mode prior to the masking are equally imprinted in the proton beam profile displayed in Fig. 2c. These intensity modulations are a common problem in high power laser beam profiles, originating for

⁷ **Obst, L. et al.** Efficient laser-driven proton acceleration from cylindrical and planar cryogenic Hydrogen jets, *Sci. Rep.* 7, doi: 10.1038/s41598-017-10589-3 (2017)

⁸ **Hilz, P. et al.** Isolated proton bunch acceleration by a petawatt laser pulse, *Nat. Commun.* 9, doi: 10.1038/s41467-017-02663-1 (2018)

⁹ **Kluge, T. et al.** High proton energies from cone targets: electron acceleration mechanisms, *New Journal of Physics* 14, doi: 10.1088/1367-2630/14/2/023038 (2012)

example in the amplification process. Especially when using large crystals as gain medium, inhomogeneities in doping, wave front and polarization properties can occur, which naturally would affect the amplified beam profile. Other candidates are inhomogeneous mirror coatings, B-integral formation in transmission optics and the granularity of compressor gratings.

While this is more of a technical than a conceptual issue, it should be possible to suppress these contributions in the laser imprint by tweaking the size of the spatial filter, i.e. target density distribution in the focus and, therefore, the spatial frequency range dominating the structure of the imprints. This will be addressed in future studies, which will be concerned with improving the concept for applications.

R: 3. In simulations, H2 is used instead of N2 in the experiment. The consequences of this choice should be more discussed. With N2, I would expect the ionization level, and hence the electron density to depend more strongly on the laser intensity, and hence to vary with z . Anyway and for any gas, the strong variations of the laser intensity (about 5 order of magnitude over 10 mm), should lead to large variations of the density and/or the temperature. The fact that the plasma field remains almost constant over ~ 4 mm in Suppl. Fig. 1b is thus unexpected and should be explained.

A: We performed 2D PIC simulations with Hydrogen as background gas and a simplified laser intensity modulation to investigate the resulting electron density and electric field distribution in this very low density and low intensity regime along region II. We also simulated one case with Nitrogen, resulting in comparable maximum electric field strengths as with Hydrogen. Naturally when using Nitrogen, the ionization state and therefore the total number of free electrons varies along the propagation axis z , and depends on the total amount of light transmitted at the target in a certain high transmission area. It has to be noted that we aim to investigate electric fields in region II, i.e. at distances sufficiently far behind the focus $z \geq 500\mu\text{m}$, where the intensity does not decrease as fast with z as estimated by the reviewer:

There, well after the focus region, the intensity decline depends on the laser beam divergence, i.e. the focal length of the focusing optic used, when propagating on macroscopic distances away from the focus. In our case of an $F2.5$ off-axis parabolic mirror, the intensity variation over 10 mm is only two orders of magnitude, i.e. roughly one order of magnitude in units of a_0 (ref. to revised main manuscript lines 122-124).

According to ionization due to the laser appearance intensity, this results in Nitrogen ionization states between $Z_i = 1$ and $Z_i = 4$ over the course of region II. Assuming an exponential lateral electron density gradient after the free electrons are heated by the laser, the electric field induced by charge separation scales¹⁰ with $\sqrt{n_e}$. Therefore, only a factor two variation in the maximum field strength is expected over the course of region II along z . Due to this comparably slow decline in electric field strength and for the sake of simplicity, we used an averaged electron density for all simulations along z , approximately resembling 3-4 fold ionization of Nitrogen. The moderate change in intensity over macroscopic distances also results in a gradual decline in electron temperature, which roughly scales with a_0 to a_0^2 (low intensity limit). As the electric field scales with $\sqrt{k_B T_e}$, this leads us to expect a change in electric field of roughly one order of magnitude over 10 mm for $z \geq 500\mu\text{m}$.

Figure 1 of this letter visualizes the decline in electric field strength along z calculated according to [8] for Nitrogen and Hydrogen, at pressure and density settings realized in experiment and PIC simulation. Absolute calculated values may vary compared to simulation results, as the Debye-length resulting from simulations turned out to be longer than predicted by the analytical calculations performed here. Nevertheless, field strengths derived for Nitrogen at 1.6×10^{-3} mbar pressure (experimental conditions) are fairly comparable to those for ionized Hydrogen at an electron density $n_e = 10^{14} \text{ cm}^{-3}$ (PIC simulation setting), as well as Nitrogen at 2.3×10^{-5} mbar to Hydrogen at $n_e = 10^{12} \text{ cm}^{-3}$.

¹⁰ Mora P. Plasma Expansion into a Vacuum, *Phys. Rev. Lett.* 90, doi: 10.1103/PhysRevLett.90.185002 (2003)

Figure 1) Decline of maximum electric field strength according to [8] for Nitrogen (solid lines) and ionized Hydrogen (dashed lines) at two different vacuum pressures realized in the experiment and density settings in the simulation, respectively. Steps in the course of the electric field for Nitrogen are the result of a change in ionisation state.

The moderate decline of the generated electric fields is confirmed in our simulations, depicted in Suppl. Fig. 2b) of the revised supplementary material.

In the experiment with the cryogenic Hydrogen jet as laser target, the residual gas in the vacuum chamber indeed consisted primarily of Hydrogen gas. However, the vacuum chamber pressure, in particular around the target, could not be controlled and measured as accurately as when deploying a precision air valve with the tungsten wire targets. Therefore, the simulation is oriented along the two vacuum chamber pressures realized in the tungsten wire experiment with air as residual gas.

We substantially extended the section Course of the electric field strength along the proton track of the supplementary material to incorporate the Figure 1 of this answer letter, as well as the reasoning laid out here, to explain the analytical and simulation results of the electric field strength (refer to the Marked_Up_Supplementaries.pdf document for all marked changes).

We also changed the first sentence of the 2D-PIC simulations of E_{trans} in low density plasmas-section of our revised manuscript from

“Two-dimensional PIC simulations were performed to investigate the emergence and evolution of electro-static fields over the course of tens of picoseconds after the propagation of a laser pulse through residual gas.”

to

“Two-dimensional PIC simulations were performed to investigate the emergence and evolution of electro-static fields over the course of tens of picoseconds after the propagation of a laser pulse through residual gas in region II.” (page 4 lines 153 ff. of the revised manuscript), to emphasize that the PIC simulations are concerned with the deflecting fields in region II.

R: Maybe I misunderstood this part, but it seems also to me that a few of hundred of eV temperature is not consistent with three to four-fold ionization of Nitrogen molecules (electrons with energy > 98eV would lead to five-fold ionization).

A: The reviewer is right in pointing out that collisional ionization could play a role at the given electron temperatures that are in the order of 10 to few 100 eV according to our simulations. This would lead to higher ionization states of Nitrogen than calculated from the appearance intensity describing barrier suppression ionization. However, at a Nitrogen gas density of $\sim 10^{10} \text{ g cm}^{-3}$, corresponding to a chamber pressure of $2.3 \times 10^{-3} \text{ mbar}$, electron-ion collision rates are only in the order of 0.1 to 10 per millisecond (¹¹ and time-resolved FLYCHK simulation displayed in Fig. 2 of this letter). As the described phenomena occur within picoseconds to

¹¹ Dendy, R. Plasma physics: an introductory course. Cambridge University Press (1993)

nanoseconds after laser-target interaction, collisional ionization of residual gas molecules can be omitted.

The following sentence was added to the supplementary material to justify omitting collisional ionization in simulations: “Collisional ionization plays no role in this very low density regime, as electron-ion collisions only occur at a maximum rate of 0.1 to 10 per millisecond at the given electron temperatures of 10 to few 100 eV.” (page 2 lines 52-53 in revised suppl. material)

Figure 2) Time-resolved FLYCHK calculation of collisional ionization states at an electron temperature of 100 eV.

R: 4. In the wire experiment the wire diameter (10 μm) is large compared to the laser beam (3 μm FWHM). The percentage of the laser energy going around the target should therefore be much smaller than in the Hydrogen jet experiment. This may be commented.

A: The amount of transmitted light in the case of the 10 μm diameter tungsten wire is 30-40%, i.e. roughly a factor 2 lower than when using the 5 μm cylindrical jet. So far this information was not given in our manuscript, so we changed a sentence in the description of the tungsten wire experiment (see our answer to comment 1 of Reviewer #1 for details).

R: The fact that 70% of the energy passes around the target in the case of the Hydrogen target is also a bit weird. For a FWHM of 3 μm , 86% of the laser energy is expected to be in a diameter of 5,1 μm .

A: As mentioned in our answer to comment 1 of Reviewer #1, the amount of transmitted light is deduced from imaging a ceramic screen positioned downstream of the laser-target interaction which results in 70% transmission in the case of the Hydrogen target. In our original manuscript we had mistakenly stated that this agrees with the result of subtracting a target-shaped obstacle from our high-dynamic range images of the focus. We corrected this error in the current version of the manuscript.

It is in fact quite surprising to observe such high transmission values in our experiments.

Naturally, the numbers stated by the reviewer (86% within 5.1 μm diameter which is where the laser intensity has dropped to $1/e^2$) apply to perfect gaussian beams, while a realistic laser

focus might contain significantly less energy within its $1/e^2$ diameter¹². Moreover, it is very challenging to measure a high-power focus (we show a low-power focus image in our manuscript) and it is therefore not clear, how well confined the laser energy is to the focal spot on the actual high intensity laser shot on target.

However, the amount of transmitted light, as long as it ranges in the few 10%, only defines the length of the ionized residual gas area behind the target along z. This results from the fact that the decline in intensity along z is rather slow, as explained in our answer to question 3 of the reviewer. Therefore, even if only 10% of the transmitted light was strongly spatially filtered at the target, the resulting low-density plasma columns would still extend up to a distance of ~ 6 mm behind the target.

R: 5. More details should be given on the Zemax simulation. Page 2, ray-tracing simulations are mentioned. Such simulations does not seem appropriate for Fourier filtering.

A: We used the „physical optics propagation“ option in Zemax which applies scalar diffraction theory to simulate the propagation of an electric field through free space and around obstacles. To clarify this in the manuscript, we replaced the following sentence:

“Comparison of the transmitted light profile with ray-tracing simulations of the inverted filter confirm the above interpretation (see Zemax calculation displayed in Fig. 1)”

with

“This was confirmed in field propagation simulations that apply scalar diffraction theory to simulate the propagation of the laser field through space, in this case around obstacles in the collimated beam and the inverted filter in the laser focus (see Zemax calculation displayed in Fig. 1).” (page 2 lines 57 ff.)

R: 6. The control of the beam divergence and direction may be challenging with existing techniques but they are not clearly demonstrated in the paper. More caution in the introduction would be appreciate.

A: We agree that the wording in the introduction was not clearly distinguishing proton beam structuring from manipulating proton beam divergence and direction. Therefore, we made the following changes to abstract and introduction of our manuscript:

Abstract:

- page 1 line 16 ff.: *“So far, only complex micro-engineering of the relativistic laser-plasma accelerator itself and limited adoption of conventional beam optics provided access to global beam parameters, such as direction and divergence.”*
was changed to: *“So far, only complex micro-engineering of the relativistic laser-plasma accelerator itself and limited adoption of conventional beam optics provided access to beam parameters that define propagation.”*

Introduction:

- please refer to our answer to the reviewer’s comment concerning practical applications of the presented scheme (at the beginning of the reviewer’s report). By including the sentence *“These approaches, while key to control proton beam propagation, are often difficult to realize in application-oriented experiments, and, more importantly, do not allow tailoring of structures on the generated beam profile according to a specific design.”*, we aim to differentiate more clearly between existing techniques that control beam divergence and direction and our new approach that structures the beam profile.

R: Further, the term “model” in the first part seems a bit excessive. I would rather use “scenario”, “analysis”, or “interpretation”.

¹² Nakamura, K. *et al.* Diagnostics, Control and Performance Parameters for the BELLA High Repetition Rate Petawatt Class Laser, *IEEE Journal of Quantum Electronics* 53, doi:10.1109/JQE.2017.2708601 (2017); Pirozhkov, A. *et al.* Approaching the diffraction-limited, bandwidth-limited Petawatt, *Optics Express* 25, doi:10.1364/OE.25.020486 (2017); Hartmann, J. *et al.* The spatial contrast challenge for intense laser-plasma experiments, *Journal of Physics: Conference Series* 1079, doi:10.1088/1742-6596/1079/1/012003 (2018)

An: We changed the word “model” to “scheme” or “scenario”, where it occurred in the manuscript and supplementary material.

Reviewer #3

Reviewer: The manuscript “All-optical structuring of laser-driven proton profiles” by L. Obst et al. reports on proton radiography of laser-induced electric fields in an underdense plasma. Proton radiography is a well-established technique in the field, especially to measure electric or magnetic field structures generated by intense lasers in plasmas (e.g. Borghesi PoP 9, 2214 (2002), PRL 92, 055003 (2004), [6,7]) and more relevant for this manuscript, to visualize the charge-displacement channel formation dynamics following relativistic self-focusing of laser pulses (Kar NJP 9 402 (2007), Willingale PRL 106, 105002 (2011)). In the present manuscript, such a pump probe setup was realized with a single focused laser pulse. The central part of the laser pulse in the focal plane was used to accelerate the proton beam from a thin cylindrical H₂ jet/ wire target. The outer parts of the laser (missing the wire) ionized background gas downstream of the target which induced electric fields that in turn are probed by the slower proton beam. Objects are then inserted in the outer region of the near field of the laser which visualize in the proton profiles, because the protons can propagate undisturbed in the absence of ionizing laser light in the low density gas. Such an experiment configuration is very common (see above references) with the neat exception that the authors realize this with a single laser pulse. The electric fields probed are also not exceptional in structure or magnitude.

In view of the above, novelty and significance of this work are limited and only represent an incremental advancement of the field. The authors’ argumentation of “direct control of the proton profile” or “novel all-optical concept to modulate the profile of the proton beam” is misleading and they appear to be overselling their claims. In conclusion, I cannot recommend publication in Nature Communication.

Answer: We thank Reviewer #3 for evaluating our work. We agree that so far our manuscript did not clearly provide distinction between our experimental arrangement and past studies utilizing laser-driven protons for the measurement of transient electric fields accompanying plasma phenomena. Please refer to the end of our answer for the changes that we made to our manuscript for clarification.

However, focusing on the proton probing aspect of our manuscript is beside the point as the main claims of our work are as follows:

- 1) The discovery of quasi-static electric fields that are formed due to laser light leaking around a target in laser-proton acceleration experiments
- 2) The fact that the generated field strengths at common vacuum conditions are high enough for a measurable proton deflection
- 3) The target for laser-proton acceleration acts as a tunable spatial frequency filter for the initial laser intensity mode, thereby selecting features that are then imprinted in the proton beam

Particularly the combination of these three observations is certainly new, as acknowledged by reviewers #1 and #2. Also, to our knowledge, it is so far not mentioned in the literature. Feedback that we received when presenting these results at meetings and conferences indicate that the community is not aware of this fact.

The significance of this discovery is twofold and we would like to refer to and elaborate on our answer to the comment of Reviewer #2, inquiring about the applicability of our scheme (page 2 of this letter). First, all-optical structuring of proton beams to a dedicated target design is so far not achieved in a simple setup as the one presented here. The works cited by the reviewer¹³

¹³ Kar, S. et al. Dynamics of charge-displacement channeling in intense laser–plasma interactions, *New Journal of Physics* 9, doi:10.1088/1367-2630/9/11/402 (2007); Willingale, L. et al. High-power, kilojoule class laser channeling in millimeter-scale underdense plasma, *Phys. Rev. Lett.* 106, doi:10.1103/PhysRevLett.106.105002 (2011)

refer to highly complex experiments at orders of magnitude higher gas densities and laser intensities, requiring two laser beams and challenging target manufacturing. Preparing laser-driven proton acceleration for applications requires simple and compact setups that allow for low-cost and high-repetition rate laser-target assemblies.

Second, the impact on the interpretation of proton beam modulations, investigated and presented multiple times in high-ranking journals in the context of plasma filamentation instabilities occurring at the target, is game changing (see references in the Discussion and outlook section of our manuscript).

Our results might even give new momentum to the radiation pressure acceleration research in the “light sail” (RPA-LS) regime, which requires the use of ultra-thin targets with the aim of accelerating the entire target foil. Optimizing the onset of transparency is the subject of ongoing studies¹⁴. Naturally, experimental studies into this subject observe a large amount of transmitted light reaching the half space behind the laser target. While observed proton beam modulations were often attributed to plasma instabilities in the target, our results offer an alternative explanation including easy control via the improvement of given vacuum conditions. Not only the latter experimental studies potentially have to be revisited, but also those utilizing laser-proton sources for the above mentioned radiography measurements in case significant amounts of laser light could have propagated around or through the proton source target.

To summarize, whenever modulated proton beam profiles will be investigated in the future, experiments will have to be conducted with a clear awareness of vacuum conditions and the alignment of transmitted laser light with respect to the proton propagation axis.

We made the following changes to the introduction of our manuscript along the lines of this reasoning:

- we changed (page 1 lines 29 ff. of revised manuscript¹⁵) “*Time-resolved sampling of transient plasma phenomena with internal and external proton probes, materials and warm dense matter research, ...*”
to
“*Time-resolved radiography of transient plasma phenomena with internal and external proton probes, materials and warm dense matter research, ...*”
where we included the reference [Borghesi Phys. Plasmas (2002)] suggested by the reviewer (references are not listed here for clarity, refer to revised manuscript for all references)
- we further included (page 1 lines 52 ff. of the revised manuscript) “*While proton radiography is primarily used to probe transient electro-magnetic field structures, by definition it results in a structured proton beam. However, those experimental arrangements are generally complex as they often include at least two laser beams as well as two interaction targets, one to provide the proton probe and the other to generate sufficiently high electric fields in a high density plasma.*” where we referenced [Kar NJP 9 402 (2007), Willingale PRL 106, 105002 (2011)] suggested by the reviewer
- and (page 2 lines 59 ff. of the revised manuscript) “*Field maps induced by the TNSA drive laser itself in the residual gas of the interaction chamber are inscribed on the TNSA protons, as they probe these fields in a proton-radiography-like manner.*”
- in the caption of Fig. 2 of the revised manuscript we changed “*In the quasi-neutral region II, transverse quasi-static electric fields between the plasma electrons and the*

¹⁴ Henig, A. *et al.* Enhanced laser-driven ion acceleration in the relativistic transparency regime, *Phys. Rev. Lett.* 103, doi:10.1103/PhysRevLett.103.045002 (2009); Palaniyappan, S. *et al.* Dynamics of relativistic transparency and optical shuttering in expanding overdense plasmas, *Nature Physics* 8, doi:10.1038/nphys2390 (2012); Palmer, C. *et al.* Rayleigh-Taylor Instability of an Ultrathin Foil Accelerated by the Radiation Pressure of an Intense Laser, *Phys. Rev. Lett.* 108, doi:10.1103/PhysRevLett.108.225002 (2012); Gonzalez-Izquierdo B. *et al.* Optically controlled dense current structures driven by relativistic plasma aperture-induced diffraction, *Nature Physics* 12, doi:10.1038/nphys3613 (2016); Higginson A. *et al.* Near-100 MeV protons via a laser-driven transparency-enhanced hybrid acceleration scheme, *Nat. Commun.* 9, doi:10.1038/s41467-018-03063-9 (2018)

¹⁵ Throughout this answer letter, references to page and line numbers will apply to the document Marked_Up_Manuscript.pdf where we highlighted changes made to the originally submitted manuscript. For readability, we also submitted the Main_manuscript_rev.pdf as the revised manuscript, where changes are not marked anymore.

remaining fixed ions result in the deflection of TNSA-protons (green) accelerated in the laser focus.”

to

“In the quasi-neutral region II, transverse quasi-static electric fields between the plasma electrons and the remaining fixed ions are visualized via radiography with the accelerated protons as intrinsic probe.”

As we stated in the above mentioned answer to the comment of Reviewer #2, we included several information concerning applicability and significance of our work in the revised manuscript. We believe that potential applications lie in radio-oncology¹⁶, ion implantation¹⁷, stress testing of materials¹⁸, or for isochoric heating in warm dense matter research¹⁹. Specifically in tumor treatment, localized proton irradiation could be realized with all-optical proton beam structuring to spare healthy tissue, while inserting metal masks in the proton beam would result in undesired secondary radiation.

However, we do not claim to present a “proto-type” setup dedicated to a certain application. The goal of this work is to promote a new technique to substructure laser-driven proton beams in comparably simple setups, not least to eventually receive inspiration from other communities, such as material or health science, concerning operation parameters that would be necessary to answer the needs of a certain application. Future studies could then be concerned with demonstrating effectiveness of this technique to a certain field of research.

¹⁶ **Ledingham, K. W. D. & Galster, W.** Laser-driven particle and photon beams and some applications. *New Journal of Physics* 12, doi:10.1088/1367-2630/12/4/045005 (2010); **Kraft, S. D. et al.** Dose-dependent biological damage of tumour cells by laser-accelerated proton beams, *New Journal of Physics* 12, doi: 10.1088/1367-2630/12/8/085003 (2010); **Malka V. et al.**

Practicability of proton therapy using compact laser systems, *Phys. Rev. Lett.* 31(6), doi: 10.1118/1.1747751 (2004)

¹⁷ **Boody, F. P. et al.** Laser-driven ion source for reduced-cost implantation of metal ions for strong reduction of dry friction and increased durability, *Laser Part. Beams* 14 doi:10.1017/S0263034600010132 (1996); **Torrisi, L. et al.** Implantation of ions produced by the use of high power iodine laser, *Appl. Surf. Sci.* 217, doi: 10.1016/S0169-4332(03)00551-8 (2003)

¹⁸ **Barberio, M. et al.** Laser-accelerated particle beams for stress testing of materials, *Nat. Commun.* 9 doi: 10.1038/s41467-017-02675-x (2018)

¹⁹ **Patel, P. et al.** Isochoric heating of solid-density matter with an ultrafast proton beam, *Phys. Rev. Lett.* 91, doi: 10.1103/PhysRevLett.91.125004 (2003)

REVIEWERS' COMMENTS:

Reviewer #1 (Remarks to the Author):

Having considered the authors' response to my minor comments, I am satisfied with the response and happy to confirm my prior recommendation to publish this manuscript in Nature Communications. Additionally, I believe the point they make in the response on the need to consider possible laser propagation effects ahead of the proton beam (e.g. in semitransparent targets) - and possible structure imprint after the acceleration - is a very important one, which may lead to reconsidering prior interpretations and stimulate new investigations.

Reviewer #2 (Remarks to the Author):

The paper has been improved but my doubts on potential applications have not been removed. The results are appealing, but they might never have practical use. Yet, this is case of many papers and I agree that the results may facilitate the interpretation of future experiments. The paper should thus be of high interest to people working on laser-plasma acceleration of ions. In the end, I think the balance is positive and I recommend publication with minor corrections.

Technical comments:

- I think it would be interesting to discuss in more detail, in the paper, the assumption that the electrostatic field is nearly constant in region II. I agree that my estimate of a decrease by 5 orders of magnitude of the laser intensity was an upper value. Yet, I think that a distance of 150 μm is already well beyond the Fresnel region (the Rayleigh length is $z_r \sim 25 \mu\text{m}$). The region II corresponds thus to $z \sim 0.15 - 15 \text{mm}$. The laser intensity I varies as z^2/z_r^2 for $z \gg z_r$. It follows that the intensity decreases by 4 orders of magnitude between $z=0.15 \text{ mm}$ and $z=15 \text{ mm}$. Because of this large decrease of I , the reader cannot expect the static field to remain constant over region II. Maybe the effect of the static field over the first mm is negligible, but this should be explained.

- I realized that the structuring process studied in the paper requires the laser divergence to approximately match the proton beam divergence, otherwise the ion beam would not see the same static field as it propagates through region II, leading to a blurring of the structuring fields. This point should be discussed in the paper. It may be an important shortcoming of the proposed technique.

Answers to Reviewers

Reviewer #1:

R: Having considered the authors' response to my minor comments, I am satisfied with the response and happy to confirm my prior recommendation to publish this manuscript in Nature Communications. Additionally, I believe the point they make in the response on the need to consider possible laser propagation effects ahead of the proton beam (e.g. in semitransparent targets) - and possible structure imprint after the acceleration - is a very important one, which may lead to reconsidering prior interpretations and stimulate new investigations.

Answer: We thank Reviewer #1 for his/her confirmed recommendation for publication of our manuscript in Nature Communications.

Reviewer #2:

R: The paper has been improved but my doubts on potential applications have not been removed. The results are appealing, but they might never have practical use. Yet, this is case of many papers and I agree that the results may facilitate the interpretation of future experiments. The paper should thus be of high interest to people working on laser-plasma acceleration of ions. In the end, I think the balance is positive and I recommend publication with minor corrections.

Answer: We thank Reviewer #2 for his/her evaluation of our revised manuscript and his/her recommendation for publication in Nature Communications.

R: Technical comments:

- I think it would be interesting to discuss in more detail, in the paper, the assumption that the electrostatic field is nearly constant in region II. I agree that my estimate of a decrease by 5 orders of magnitude of the laser intensity was an upper value. Yet, I think that a distance of 150 μm is already well beyond the Fresnel region (the Rayleigh length is $z_r \sim 25 \mu\text{m}$). The region II corresponds thus to $z \sim 0.15 - 15 \text{mm}$. The laser intensity I varies as z^{-2}/z_r^{-2} for $z \gg z_r$. It follows that the intensity decreases by 4 orders of magnitude between $z=0.15 \text{mm}$ and $z=15 \text{mm}$. Because of this large decrease of I , the reader cannot expect the static field to remain constant over region II. Maybe the effect of the static field over the first mm is negligible, but this should be explained.

A: As we describe in our manuscript, the lower limit of region II is estimated heuristically by stating that the laser intensity profile must have regained the so-called "near-field features", i.e. the shape of the intensity profile of the collimated laser beam. We further state that this happens well beyond the Fresnel region, after few hundred μm behind the focus, as verified during the experiment (ref. to page 3 lines 121 ff. of the revised manuscript). Figure 1 of this Answer letter displays the laser intensity profile at a distance of 300 μm from the laser focus, recorded during the experimental campaign. Clearly, its shape does not resemble that of the all-optically shaped proton beam profile. We therefore conclude that region II starts at larger distances from the focus. While in region I and the transition area the laser might still establish electric fields in the residual chamber gas, our experimental results show that these fields

clearly do not dominate the resulting proton beam profile in the way fields in region II do. The detailed effects of electrostatic fields in region I on the resulting proton beam profile are still subject to investigation, while we started discussing some effects in Supplementary Note 1. As we state in Supplementary Note 2, we determine the region II in our experimental configuration to start at roughly 500 μm behind the focus. A moderate (\sim one order of magnitude over 10 mm) decline in electric field strength after 500 μm behind the laser focus is derived from the analytical description, and confirmed by Particle in Cell simulations. It is based on this moderate decline in electric field that we make a simplified example calculation in our manuscript, assuming (only for this example) that the electric field strength along region II is constant. The result of this example calculation is an estimate of the expected field strength and is given very roughly, as ranging from 10^6 to 10^7 Vm^{-1} .

Figure 1: Laser intensity profile at 300 μm distance from the laser focus.

The distances from the focus where region I transits to region II and region II to region III are highly dependent on laser- and target configurations. The values that we present in our manuscript therefore apply to our experimental conditions but might vary in other experiments. Due to the macroscopic length of our observed region II of >10 mm, these variations are not expected to be of great impact for the more general results of laser-proton imprinting.

We made the following changes to our manuscript for clarification:

- We changed
The lower boundary, i.e. the transition to region I is estimated more heuristically: acknowledging that characteristic structures in the laser near-field [...]
to
The lower boundary, marking the transition to region I, while subject to ongoing investigations, is estimated heuristically: acknowledging that characteristic structures in the laser near-field [...]
(page 3 lines 119 ff. of the revised manuscript)
- We changed
As verified during the experiment, this only happens at distances larger than few hundred μm behind the focus, resulting in an approximate length of region II of 14-15 mm (refer to supplementary material for further discussion).
to
As verified during the experiment, this only happens at a distance of roughly 500 μm behind the focus, resulting in an approximate length of region II of 14-15 mm (refer to Supplementary Notes 1 and 2 and Supplementary Figure 1 for further discussion).
(page 3 lines 123 ff. of the revised manuscript)
- We changed

An estimate for the field strength E_{trans} is derived from the experimentally recorded proton beam profiles.

to

A first estimate for the field strength E_{trans} is derived from the experimentally recorded proton beam profiles in the form of an example calculation.

(page 3 lines 130 ff. of the revised manuscript)

And in the supplementary material:

- We changed

Features that remain smaller than $2 \times \lambda_D$ over the entire ionized region ($z \sim 15$ mm given our laser conditions) are therefore not expected to reappear in the proton beam profile.

to

As will need to be confirmed in experiments, features that remain smaller than $2 \times \lambda_D$ over the entire ionized region (with a length of 14 - 15 mm for our laser conditions) therefore should not reappear in the proton beam profile.

(page 1 lines 38 ff. of the revised supplementary material)

R:

- I realized that the structuring process studied in the paper requires the laser divergence to approximately match the proton beam divergence, otherwise the ion beam would not see the same static field as it propagates through region II, leading to a blurring of the structuring fields. This point should be discussed in the paper. It may be an important shortcoming of the proposed technique.

A: This is usually the case for TNSA (and other laser-ion acceleration mechanisms proposed in the literature), and is therefore inherently assumed in our manuscript. The case where laser- and proton beam divergences are significantly different will need to be assessed in future studies. It might well be that over long deflection lengths and sufficiently high deflection fields, an initial offset in alignment between proton trajectory and low density plasma column evens out.

We added the following sentence to our manuscript to clarify the assumption of sufficiently similar laser- and proton beam divergence:

- *In the framework of laser-proton acceleration it can be assumed that the proton and laser beam divergence in the ionized areas in region II are sufficiently similar such that a proton experiences the deflecting fields of one low-density plasma column over the length of region II, leading to the observed sharp features in the final proton beam profile.*

(page 3 lines 113 ff. in the revised manuscript)